# Equilibrium Phase Relations for a SiO$_2$-Al$_2$O$_3$-FeO$_x$ System at 1300 °C and 1400 °C in Air

**Song Li [1,2], Yuchao Qiu [3,4], Junjie Shi [3,4],\*, Jianzhong Li [3,4] and Changsheng Liu [5]**

[1]  School of Chemistry and Materials Engineering, Liupanshui Normal University, Liupanshui 553004, China; lpssylis@126.com
[2]  Guizhou Provincial Key Laboratory of Coal Clean Utilization, Liupanshu 553004, China
[3]  Key Laboratory for Ecological Metallurgy of Multimetallic Mineral (Ministry of Education), Northeastern University, Shenyang 110819, China; qiuyc2004@126.com (Y.Q.); lijz@mail.neu.edu.cn (J.L.)
[4]  School of Metallurgy, Northeastern University, Shenyang 110819, China
[5]  School of Material Science and Engineering, Northeastern University, Shenyang 110819, China; csliu@mail.neu.edu.cn
\*  Correspondence: junjieshi@126.com

**Abstract:** A long-term fundamental study for the construction of the thermodynamic database of a metallurgical slag system has been proposed. In the present work, the equilibrium phase relations for the key ternary SiO$_2$-Al$_2$O$_3$-FeO$_x$ system at 1300 °C and 1400 °C in air were experimentally determined by the equilibrium-quenching technique, followed by X-ray Photoelectron Spectroscopy and Scanning Electron Microscope equipped with an Energy Dispersive X-ray Spectrometer analysis. The oxidation states of Fe$_2$O$_3$ and Fe$_3$O$_4$ were confirmed at 1300 °C and 1400 °C, respectively, from both XPS detection and FactSage calculation. Within the high-SiO$_2$ composition range, the solid phases of silica, mullite, magnetite and ferric oxide were confirmed as the equilibrium phases. Based on the equilibrium compositions, the 1300 °C and 1400 °C isotherms were projected onto a SiO$_2$-Al$_2$O$_3$-FeO$_x$ quasi-ternary phase diagram; however, obvious discrepancies with about a 20 °C difference were confirmed from further comparison with the predictions given by FactSage, indicating that more efforts are needed for the updating of the current thermodynamic database relating to metallurgical slag oxide systems.

**Keywords:** slag; equilibrium; thermodynamic; CALPHAD; phase relation

## 1. Introduction

The equilibrium phase relation presented by a phase diagram is of fundamental importance for the development of material and process design [1]. During the production of steel from iron ore, the processing parameters—i.e., the temperature, composition and oxygen partial pressure—are basically determined according to phase diagrams, such as the Fe$_t$O-SiO$_2$-based system for sinter production [2], the CaO-SiO$_2$-based and CaO-SiO$_2$-MgO-Al$_2$O$_3$-based systems for the design of furnace and converter smelting slag [3], the Fe-C-based systems for heat treatment [4,5], and the CaO-SiO$_2$-MgO-Al$_2$O$_3$-TiO$_2$ system for the comprehensive utilization of titania-bearing slag [6,7], etc. It is now well known that the thermodynamic properties of the slag have a significant influence on the metal–slag reaction, crystallization, inclusion formation, oxidation, and degasification, etc. [8], and the colloquial description "steel smelting is actually slag smelting" has been widely accepted in industry and research. After a long-term continuous study of the phase diagram for metallurgical oxide systems by scientists from the beginning of the 1900s, most of the phase diagrams for binary and parts of the ternary/quaternaty/high-order oxide systems relating to the high temperature range of metallurgical reactions have been extensively studied, and are mainly collected in the book 'Slag Atlas', published in 1995 [9].

However, many significant differences have been reported between early research and recent studies, and some information is presented as dashed, indicating that the accuracy of the phase relations and isotherms remains uncertain [10]. With the background of "peak carbon dioxide emission and carbon neutrality" [11], more stringent demands such as green manufacturing and energy saving have been proposed for the modern metallurgical industry for the production of steel, which requires more precise control of the operating parameters—i.e., temperature and atmosphere—according to the corresponding accurate phase diagrams [12]. Therefore, it is essential to update the phase diagram information to eliminate the discrepancies from both industrial and scientific interests.

The metallurgical slag oxide can be described basically as a $CaO$-$SiO_2$-$Al_2O_3$-$MgO$-$Fe_tO$-$MeO_x$ (Me = Ti, Cr, P, Na, K, Mn, etc) system according to different ore and smelting processes [4,13]. The industrial parameters for the iron and steel production could be easily optimized if a comprehensive physiochemical database containing key thermodynamic data for a metallurgical slag system was constructed, which is an important fundamental work depending on the long-term contribution from the scientists around the world. Until now, the thermodynamic studies for metallurgical slag systems were mainly focused on the sub-binary and parts of sub-ternary system, i.e., $SiO_2$-$Al_2O_3$ [14], $SiO_2$-$FeO$ [15], $SiO_2$-$Fe_2O_3$ [16], $SiO_2$-$TiO_2$ [17], $SiO_2$-$CaO$ [18], $SiO_2$-$Na_2O$ [19], $SiO_2$-$Al_2O_3$-$FeO$ [20], and $SiO_2$-$Al_2O_3$-$TiO_2$ [21], etc. The studies for the quaternary system and higher-order system are limited, and most of the phase diagrams collected in 'Slag Atlas' [9] for higher-order systems were mainly presented as pseudo-ternary phase diagrams with fixed compositions. In the past decades, the rapid development of computer techniques has made the construction of the thermodynamic database possible, and many pieces of commercial thermodynamic software—including FactSage [22], HSC [23,24], MTDATA [25], and Thermocalc [26]—have been constructed by the CALPHAD (CALculation of PHase Diagram) technique [27], and have been successfully used for equilibrium phase diagram predictions for oxide systems including $CaO$-$SiO_2$ [28], $Al_2O_3$-$MgO$-$FeO$ [29], and $CaO$-$MgO$-$SiO_2$-$Al_2O_3$ [30], etc. However, the calculated phase diagrams for higher-order systems are still far from accurate, and most of the calculations are only efficient within a narrow temperature and composition range. As pointed out by FactSage, the equilibrium phase relations of $TiO_x$-containing systems were mainly optimized for lower-order systems under reducing conditions [31,32], and the thermodynamic properties of the solid solution of pseudobrookite were not included in the current FactSage database. Therefore, it can be concluded that the available studies on the equilibrium phase relationships for a complex metallurgical slag system are inadequate from both experimental and modeling perspectives. The precise prediction of the physiochemical properties of slag at a high temperature requires quantitative knowledge of the thermodynamic data.

Therefore, a long-term fundamental thermodynamic investigation for the construction of the thermodynamic database of a metallurgical slag system was started a few years ago [33]. The premise of this huge project is to update the key thermodynamic properties for the key binary and ternary systems. In the present work, the core subsystem of $SiO_2$-$Al_2O_3$-$Fe_2O_3$ was selected as a prior study, and the equilibrium phase relations at 1300 °C and 1400 °C in air were extensively investigated, with particular attention being paid to the high $SiO_2$ content area. Furthermore, the comparisons of the experimental results with thermodynamic software predictions by FactSage were comprehensively conducted, in order to evaluate the need to update the current thermodynamic oxide database related to metallurgical smelting.

## 2. Experiment

The high viscosities and sluggishness of the crystallization result in the liquid phase being retained as glass during the quenching for silica-containing melts [34]; therefore, the reliable thermodynamic equilibrium-quenching technique [35] was employed to explore the equilibrium phase relations for an $SiO_2$-$Al_2O_3$-$Fe_2O_3$ system, and the experimental process could be explained by the flow chart in Figure 1.

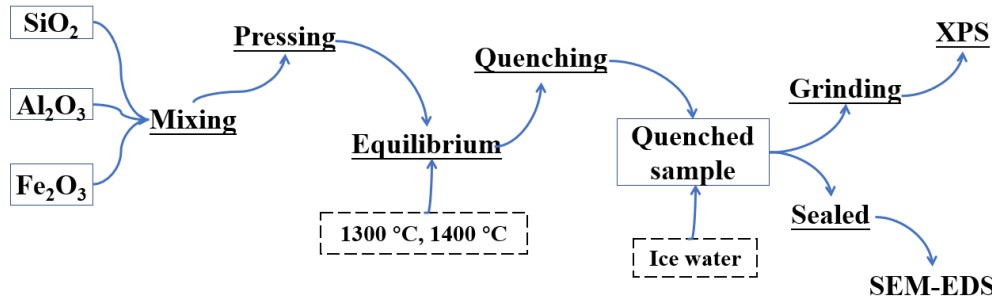

**Figure 1.** Experimental processes for the thermodynamic equilibrium-quenching technique.

High-purity oxide powders of $SiO_2$ (99.99 wt.%), $Al_2O_3$ (99.99 wt.%) and $Fe_2O_3$ (99.9 wt.%) from Sinopharm Chemical Reagent Co., Ltd., shenyang, China, were employed as the starting materials. The initial compositions were designed to reflect the influences of a high $SiO_2$ content on the equilibrium phase relations; the initial compositions for the samples are listed in Table 1 with the contents of 70 wt.% to 90 wt.% $SiO_2$, and 5 wt.% to 15 wt.% $Fe_2O_3$, while the concentration of $Al_2O_3$ was lower than 15 wt.%. Each reagent powder was weighted to an accuracy of 0.1 mg and then thoroughly mixed in an agate mortar. The mixed powders with total weight of 0.15 g were pressed into small cylinders and stored in a desiccator prior to future experiments.

**Table 1.** Equilibrium compositions for sample B1 to B6 at 1400 °C in air.

| No. | Initial Compositions, wt.% | | | Equilibrium PHASES | Equilibrium Compositions, wt.% | | |
|---|---|---|---|---|---|---|---|
| | Fe₂O₃ | SiO₂ | Al₂O₃ | | Magnetite | SiO₂ | Al₂O₃ |
| B1 | 10.0 | 90.0 | 0.0 | magnetite | 91.7 ± 1.3 | 8.0 ± 1.3 | 0.0 ± 0.0 |
| | | | | Silica | 0.5 ± 0.1 | 99.5 ± 0.1 | 0.0 ± 0.0 |
| B2 | 5.0 | 90.0 | 5.0 | Liquid | 35.0 ± 1.1 | 42.3 ± 1.1 | 22.7 ± 0.6 |
| | | | | Silica | 0.4 ± 0.2 | 99.4 ± 0.2 | 0.1 ± 0.0(5) |
| B3 | 15.0 | 80.0 | 5.0 | Liquid | 47.4 ± 1.2 | 33.0 ± 1.3 | 19.6 ± 1.1 |
| | | | | Silica | 0.7 ± 0.1 | 99.2 ± 0.1 | 0.1 ± 0.0(3) |
| B4 | 10.0 | 80.0 | 10.0 | Liquid | 29.7 ± 1.4 | 48.8 ± 1.4 | 21.5 ± 0.7 |
| | | | | Silica | 0.7 ± 0.0(9) | 99.0 ± 0.0(5) | 0.3 ± 0.1 |
| B5 | 5.0 | 80.0 | 15.0 | Liquid | 27.4 ± 0.7 | 51.5 ± 0.8 | 21.1 ± 0.4 |
| | | | | Silica | 0.4 ± 0.1 | 99.2 ± 0.1 | 0.3 ± 0.0(8) |
| | | | | Mullite | 10.0 ± 0.4 | 26.6 ± 1.1 | 63.4 ± 1.3 |
| B6 | 15.0 | 70.0 | 15.0 | Liquid | 28.0 ± 0.6 | 52.4 ± 0.7 | 19.6 ± 0.2 |
| | | | | Silica | 0.6 ± 0.1 | 99.0 ± 0.1 | 0.4 ± 0.1 |
| | | | | Mullite | 9.7 ± 0.9 | 25.6 ± 1.1 | 64.7 ± 1.4 |

The equilibrium experiments were conducted in a vertical alumina tube furnace (YeTuo Inc., Shanghai, China). Platinum foil containing the pressed cylinder sample was suspended by a platinum wire at a position within the even temperature zone of the tube in air. The temperature was monitored on-line by a B-type thermocouple with a temperature accuracy of ±3 °C, and the thermocouple was placed inside the tube just adjacent to the sample at the same height. The furnace temperature was first heated to 1500 °C to premelt the sample for 60 min; afterwards, the temperature was decreased to the equilibrium temperatures (1300 °C and 1400 °C) for a certain long time until the sample reached an equilibrium state. Preliminary experiments for silica-containing oxide systems, i.e., CaO-SiO₂ [18,28], CaO-SiO₂-TiOₓ [17,36,37], and CaO-SiO₂-MgO-TiO₂ [38], etc., proved that 1440 min was sufficient to reach the equilibrium state. Therefore, each sample was kept for at least 1440 min in the present study. The sample was rapidly dropped into ice water for quenching after it reached the equilibrium state. The purpose of the quenching process

was to maintain the phase assembly and phase composition presented at a high temperature at room temperature, which was more convenient for analysis. The basic requirement was that the temperature decrease's speed in the quenching media should be faster than the phase transformation speed that might happen during the quenching process. Therefore, ice water was selected as the quenching medium in the present work due to its excellent cooling effect. The quenched sample was then dried and stored in a desiccator for future analysis.

The powder sample was sent for X-ray Photoelectron Spectroscopy (XPS, Escalab 250Xi, Thermo Fisher Scientific, NewYork, USA) analysis of the oxidation state of the Fe element. The XPS measurement was performed with a 20 keV pass energy within an area of 500 μm. The curve fitting of the XPS spectra was carried out using XPSPEAK vers. 4.1 software (Department of Chemistry, The Chinese University of Hong Kong, Shatin, Hong Kong, China). The granular sample was mounted in epoxy resin, polished and coated with carbon, then analyzed using a Scanning Electron Microscope (SEM, ULTRA PLUS, ZEISS, Oberkochen, Germany) equipped with an Energy Dispersive X-ray Spectrometer (EDS, Thermo Fisher Scientific, Waltham, MA, USA). An accelerating voltage of 15 kV and a beam current of 10 nA on the sample surface were used for the detection. The Proza (Phi-Rho-Z) matrix correction procedure was used for the processing of the raw data. The external standards utilized in the EDS analyses were quartz (for O, K$\alpha$ and Si, K$\alpha$), Al metal (for Al, K$\alpha$), and Fe metal (for Fe, K$\alpha$). At least six analysis points were randomly selected from each phase for statistical reliability. Raymund W.M. Kwok

## 3. Results and Discussion

### 3.1. The Speciation of Fe

It was reported that the $w(FeO)/w(Fe_2O_3)$ ratio has significant influences on the formation of the equilibrium phase during smelting [39]. In order to reveal the influence of the temperature and oxygen partial pressure on the oxidation state of Fe, the predominant phase diagram calculated by FactSage 8.1 is presented in Figure 2a [40]. As can be seen, Fe can exist as $Fe_2O_3$, magnetite, FeO and Fe with the variation of the temperature and oxygen partial pressure. Under the present experimental conditions in air (as shown by points E1 and E2), the predictions indicate that Fe would be stable as $Fe_2O_3$ at 1300 °C, but it would be transformed into the magnetite ($FeO\cdot Fe_2O_3$) phase when the temperature increased to 1400 °C.

Furthermore, XPS analysis was employed to evaluate the prediction by FactSage 8.1, and to reveal the influence of other elements, i.e., Si and Al, on the oxidation state of the Fe. The XPS spectra for sample B6 at 1300 °C and 1400 °C are shown in Figures 2b and 2c, respectively. As can be seen, the Fe element was stable as trivalent $Fe^{3+}$ at 1300, while at 1400 °C, both bivalent $Fe^{2+}$ and trivalent $Fe^{3+}$ were found in sample B6. Moreover, the mol ratio of $Fe^{2+}/Fe^{3+}$ was estimated as 0.87:1 from the relative areas of the Gaussian peaks under $Fe^{2+}$ and $Fe^{3+}$ [41], which was higher than the stoichiometric mol ratio of the magnetite ($Fe_3O_4$) phase, indicating that the magnetite presented as solid solution properties, while for the consideration of convenient presentation, the stoichiometric composition of $Fe_3O_4$ was employed in the present study. The XPS analysis verified the prediction by FactSage; therefore, the oxidation states of $Fe_2O_3$ at 1300 °C and magnetite ($Fe_3O_4$) at 1400 °C were adopted for the discussion, as well as for the construction of the phase diagram in the following sections.

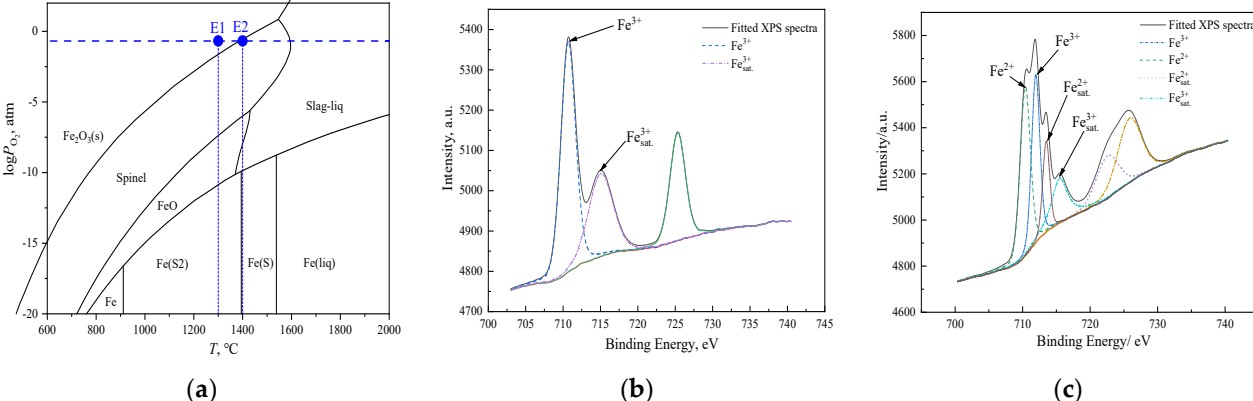

**Figure 2.** Determination of the Fe valance state in air at 1300 °C and 1400 °C. (**a**) Fe–O stable phase diagram; (**b**) Sample B6 at 1300 °C; (**c**) Sample B6 at 1400 °C.

### 3.2. Equilibrium Results at 1400 °C

The microstructure of the equilibrium phases and the equilibrium compositions at 1400 °C for samples B1 to B6 are presented in Figure 3 and Table 1, respectively. Within the studied composition range, two kinds of two-phase equilibrium and one type of three-phase equilibrium were confirmed at 1400 °C in air.

Sample B1 was $Al_2O_3$ free, and was composed of 90 wt.% $SiO_2$ and 10 wt.% $Fe_2O_3$; no liquid phase was presented at 1400 °C, and the equilibrium phase was silica and magnetite, as shown by Figure 3a. The equilibrium result of sample B1 was consistent with the $SiO_2$-$FeO_x$ binary phase diagram in air [42], as indicated in Figure 4; $Fe_2O_3$ will be transformed to $Fe_3O_4$ when temperature is above 1390 °C, showing $SiO_2$-$Fe_3O_4$ coexisting at 1400 °C. Meanwhile, for samples B2, B3 and B4, the liquid-silica two-phase equilibrium was verified (Figure 3b–d), indicating that the addition of 5 wt.% and 10 wt.% $Al_2O_3$ have a trend of decreasing the melting temperature compared with the composition of sample B1. Furthermore, the three-phase equilibrium of liquid-silica-mullite was confirmed for samples B5 and B6, as presented by Figure 3e,f. From the thermodynamic law of the phase-adjacent rule [43], it can be concluded that the primary phase fields of silica and mullite are adjacent to each other.

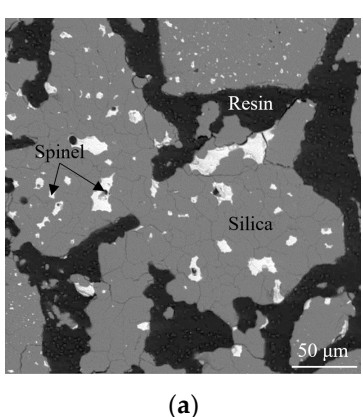

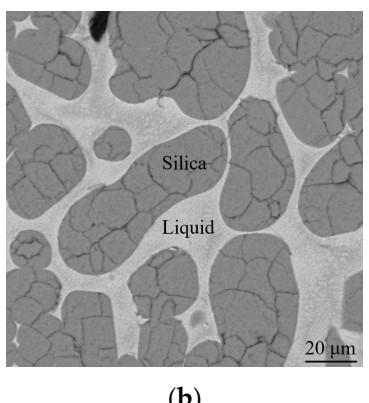

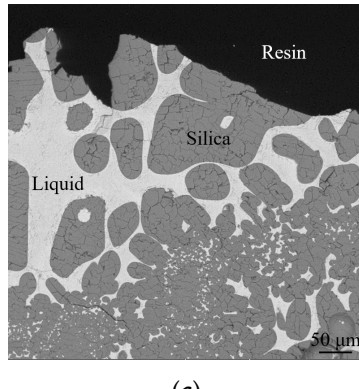

(**a**)

(**b**)

(**c**)

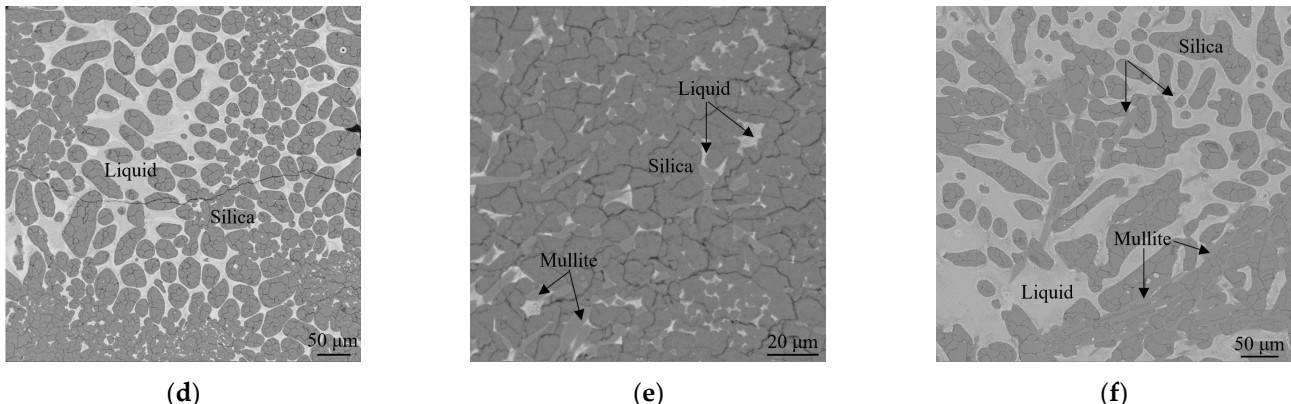

**Figure 3.** SEM micrographs for the equilibrium phases of sample B1 to B6 at 1400 °C in air. (**a**) B1 at 1400 °C; (**b**) B2 at 1400 °C; (**c**) B3 at 1400 °C; (**d**) B4 at 1400 °C; (**e**) B5 at 1400 °C; (**f**) B6 at 1400 °C.

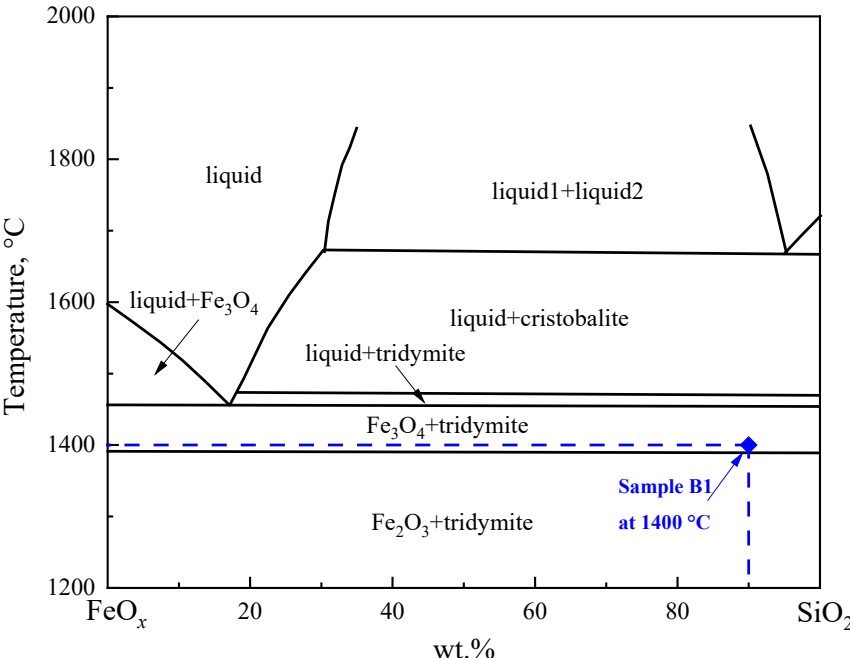

**Figure 4.** The $SiO_2$-$FeO_x$ binary phase diagram in air [42].

### 3.3. Equilibrium Results at 1300 °C and 1400 °C

The SEM micrographs and compositions for the equilibrium phases at 1300 °C are shown in Figure 5 and Table 2, respectively. The equilibrium results at 1300 °C for samples B5 and B6 are relatively straightforward compared with the results at 1400 °C. As illustrated by Figure 5a,b, both samples B5 and B6 presented as a silica-mullite-$Fe_2O_3$ three-solid-phase equilibrium, indicating that 1300 °C is below the ternary eutectic temperature. Similarly, it can be estimated that the primary phase fields of silica, mullite, and $Fe_2O_3$ are adjacent to each other, from the perspective of thermodynamic equilibrium theory [43].

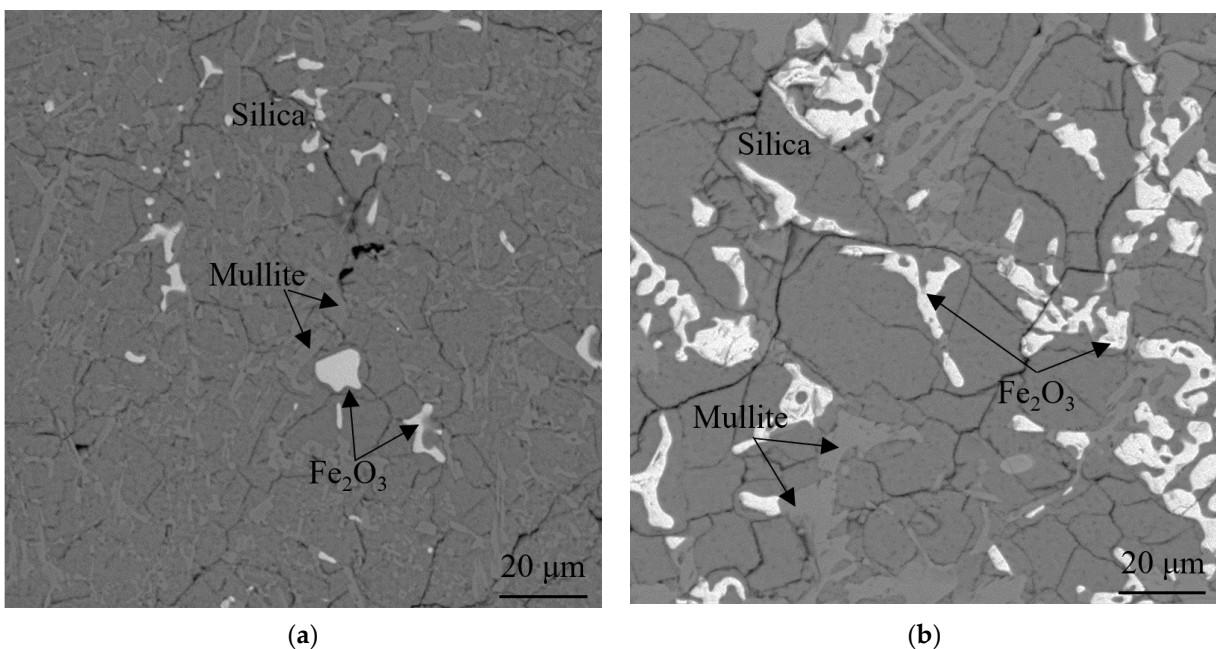

**Figure 5.** SEM micrographs for the equilibrium phases of samples B5 and B6 at 1300 °C in air. (**a**) B5 at 1300 °C; (**b**) B6 at 1300 °C.

**Table 2.** Equilibrium compositions for samples B5 and B6 at 1300 °C in air.

| No. | Initial Compositions, wt.% | | | Equilibrium Phases | Equilibrium Compositions, wt.% | | |
|---|---|---|---|---|---|---|---|
| | $Fe_2O_3$ | $SiO_2$ | $Al_2O_3$ | | $Fe_2O_3$ | $SiO_2$ | $Al_2O_3$ |
| B5 | 5.0 | 80.0 | 15.0 | Silica | 0.3 ± 0.0(9) | 99.3 ± 0.1 | 0.4 ±0.0(6) |
| | | | | Mullite | 5.1 ± 0.5 | 27.7 ± 1.1 | 67.2 ± 1.5 |
| | | | | Ferric oxide | 77.7 ± 1.3 | 7.5 ± 0.9 | 14.8 ± 1.1 |
| B6 | 15.0 | 70.0 | 15.0 | Silica | 0.6 ± 0.1 | 99.1 ± 0.1 | 0.3 ± 0.0(9) |
| | | | | Mullite | 13.6 ± 0.8 | 25.9 ± 0.4 | 60.6 ± 0.7 |
| | | | | Ferric oxide | 87.2 ± 0.5 | 0.7 ± 0.1 | 12.1 ± 0.4 |

*3.4. Projection of the 1400 °C Isotherm*

Based on the experimental results described above, the 1300 °C and 1400 °C isothermal phase diagrams were constructed and projected onto a $SiO_2$-$Al_2O_3$-$Fe_3O_4$ phase diagram in Figure 6a and 6b, respectively. In Figure 6a, the compositions for the equilibrium solid phases at 1300 °C (orange symbols) are projected; as can be seen, the compositions of the equilibrium silica phases are consistent with the stoichiometric composition, while both the equilibrium mullite and ferric oxide phases appeared to be solid solution phases. The composition of the mullite solid solution shifts to a somehow higher $Fe_2O_3$ content direction, while the composition of the ferric oxide shifts to higher $Al_2O_3$ and $SiO_2$ concentrations.

In Figure 6b, the 1400 °C isotherm calculated by FactSage 8.1 is also presented for a comprehensive comparison with the experimental results and previous literature result [38]. The prediction from FactSage 8.1 was performed by the "Equilib" module with reference to the FactPS and FToxide databases [44]. The calculated 1400 °C isotherm is shown by the dashed line, while the literature result is presented by the dash-dot-dashed line. Obvious divergence with about 20 °C was discovered from the comparison, the experiment determined that the 1400 °C isotherm moved towards much higher $Al_2O_3$ and $SiO_2$ contents. The calculated 1400 °C isotherm showed the same trend as the literature data [42]; however, the liquidus area given by FactSage 8.1 was much smaller in comparison. The discrepancies between the experiments and FactSage 8.1 indicate that more efforts are

needed to update the current thermodynamic oxides database related to the metallurgical slag oxide database, from both an industrial application and scientific research point of view.

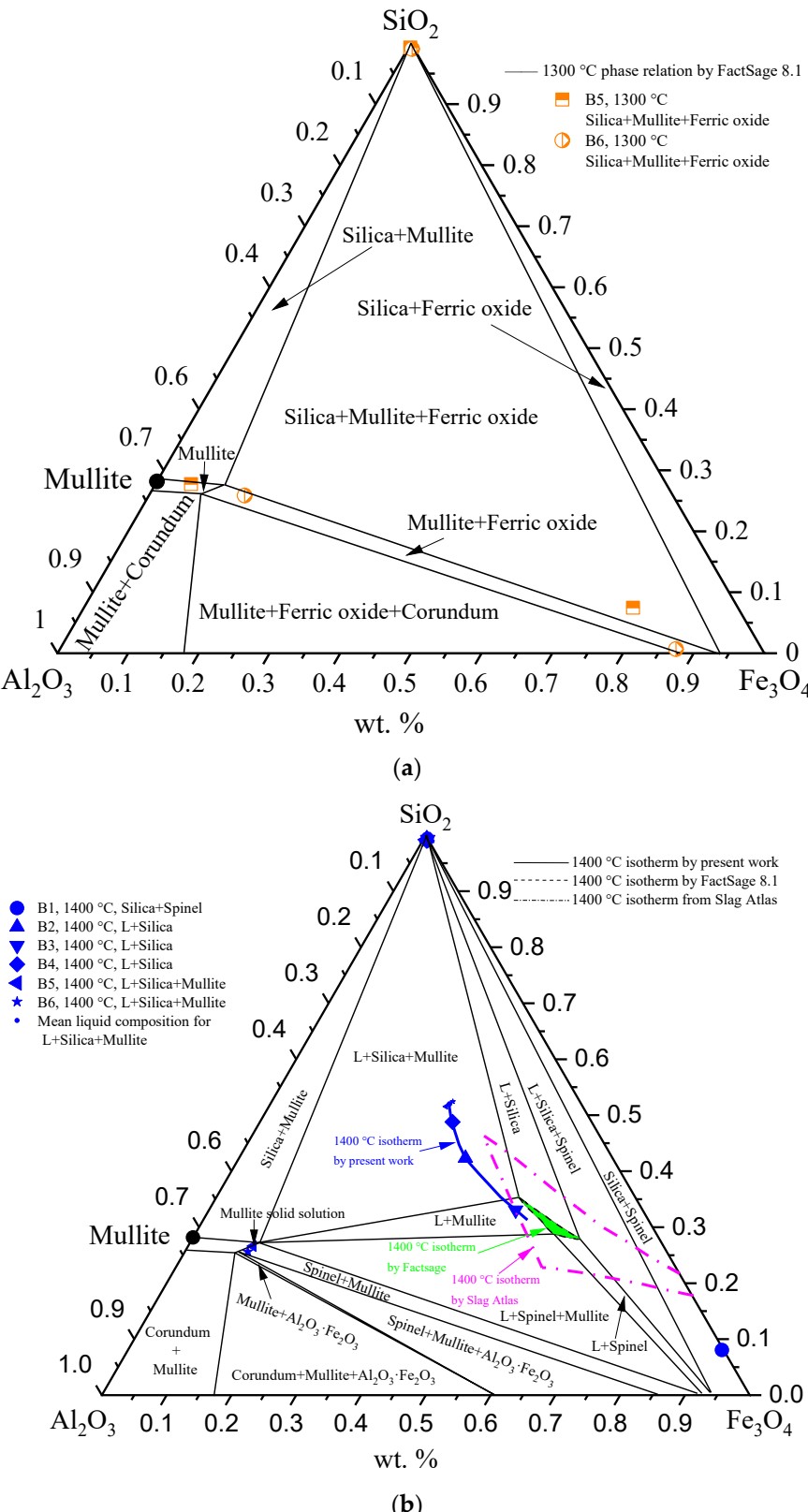

**Figure 6.** Comparison of the 1300 °C and 1400 °C isotherms from the experimental results with FactSage predictions and the literature results. (**a**) 1300 °C isotherm; (**b**) 1400 °C isotherm.

### 4. Conclusions

The thermodynamic equilibrium phase relations of the key binary and ternary subsystems are important for the development of an integrated metallurgical slag thermodynamic database. In the present work, the equilibrium-quenching technique was employed to investigate the phase relationships for the key $SiO_2$-$Al_2O_3$-$FeO_x$ system at 1300 °C and 1400 °C in air within the high-$SiO_2$-content range. The solid phase of silica and the solid solution phases of mullite, magnetite and ferric oxide were confirmed to coexist with the liquid phase. The 1300 °C and 1400 °C isotherms were then plotted on the $SiO_2$-$Al_2O_3$-$Fe_3O_4$ phase diagram based on the experimental results. Significant discrepancies were found between the experimental results and the predictions given by FactSage, showing the updating directions for the thermodynamic database related to the metallurgical slag oxide system.

**Author Contributions:** Literature search, manuscript writing, data analysis and funding support, S.L.; production of graphics and data collection, Y.Q.; Program design, macuscript writing and funding support, J.S.; program desigh, J.L.; program design and data analysis, C.L. All authors have read and agreed to the published version of the manuscript.

**Funding:** This research was funded by [China Postdoctoral Science Foundation] grant number [2020TQ0059], [2020M680967]; [The Natural Science Foundation of Liaoning Province] grant number [2021-MS-083]; [The Fundamental Research Funds for the Central Universities] grant number [N2125010], [Guizhou Provincial Key Laboratory of Coal Clean Utilization (qiankeheping-tairencai)] grant number [[2020]2001)], [The Academician Workstation of Liupanshui Normal University (qiankehepingtairencai)] grant number [[2019]5604)], and [The Key disciplines of Liupanshui Normol University] grant number [(LPSSYZDXK202001)].

**Data Availability Statement:** Not applicable.

**Conflicts of Interest:** The authors report no conflicts of interest, and the authors alone are responsible for the content and writing of the article.

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
