# Peer review of "Equilibrium Phase Relations for a SiO2-Al2O3-FeOx System at 1300 °C and 1400 °C in Air"

_metals, doi:10.3390/met12060926_

Round 1

Reviewer 1 Report

This research produces a limited number of new equilibrium data points at two temperatures in the FeOx-SiO2-Al2O3 system. The uncertainty levels of the analyzed equilibrium compositions are not presented; more importantly, there is no measurement of the actual Fe(2+)/Fe(3+) ratios in the samples equilibrated at 1400 °C. It is instead presumed that the ratio is 2:1, and that all of the iron is present as 'spinel', which may not be the case. (Note: FeO.Fe2O3 should be called magnetite, to distinguish it from other possible spinel compositions.) The results from modeling software would be more reliable than the results of this research.

Author Response

Comment: This research produces a limited number of new equilibrium data points at two temperatures in the FeOx-SiO2-Al2O3 system. The uncertainty levels of the analyzed equilibrium compositions are not presented; more importantly, there is no measurement of the actual Fe2+/Fe3+ ratios in the samples equilibrated at 1400 °C. It is instead presumed that the ratio is 2:1, and that all of the iron is present as 'spinel', which may not be the case. (Note: FeO·Fe2O3 should be called magnetite, to distinguish it from other possible spinel compositions.) The results from modeling software would be more reliable than the results of this research.

Response: Thanks for reviewer’s professional comments. In the present experiments, the compositions of the equilibrium phases were detected by the EDS, which is accurate to one decimal place, therefore, all the equilibrium compositions were presented as one decimal in Table 1 and Table 2 together with the square deviation.

It is really rigorous to have a precise measurement for the actual ratios of Fe2+/Fe3+, therefore, the mol ratio of Fe2+/Fe3+ was estimate as 0.87:1 according to the relative areas of the Gaussian peaks under Fe2+ and Fe3+ by the methods from reference [R1], the result indicate that the magnetite phase presented as solid solution phase properties, which is consistent with the information from Fe-O phase diagram. While for the consideration of convenient presentation, the stoichiometric composition of Fe3O4 was employed in the present study. For our future experiments, more attention will be paid for the analysis of the Fe2+/Fe3+ ratio. The corresponding discussion has been revised accordingly in part 3.1 of manuscript.

It is a good suggestion to unify the FeO·Fe2O3 as magnetite, therefore, spinel has been replaced as magnetite from the full text.

Reference:

R1. Liu R, Zhang Y, Duan L, et al. Effect of Fe2+/Fe3+ ratio on photocatalytic activities of Zn1-xFexO nanoparticles fabricated by the auto combustion method[J]. Ceramics International, 2020, 46(1): 1-7.

Reviewer 2 Report

The present manuscript entitled “1300 °C and 1400 °C Isotherms of SiO2-Al2O3-FeOx System in Air” by Song Li et al., describes the equilibrium phase relations for the key SiO2-Al2O3-FeOx system at 1300 °C and 1400 °C in air were experimentally determined by the equilibrium-quenching technique followed by XPS and SEM & EDX analysis. However, obvious discrepancies with about 20 °C were confirmed from further comparison with the predictions by FactSage were comprehensive conducted, to evaluate the need for updating the current thermodynamic oxide database related to metallurgical smelting.. The authors report an interesting approach but the presentation of the work is clear. The objective and justification of the work are clear, and the experimental work is significant. The study is accurate and adequate, and thus, I would recommend it for publication in Metals. However, certain Minor issues are detailed below to improve the quality of the manuscript.

I advise the authors to take the following points into account while revising their manuscript.

Comment 1: There are some typographical errors in the manuscript text, so the authors need to correct them in the revised manuscript. In the whole manuscript, the authors must be taken care of the superscripts and subscripts, and abbreviations.

Comment 2: Abstract is poorly written, it needs to be revised.

Comment 3: The authors need to add and discuss some more recent literature in the introduction section to strengthen the background of their work.

Comment 4: The authors should explore and discuss the XPS results (Section 3.1.) with some more references to prepare a better discussion.

Comment 5: The homogeneity of the reference section needs to be maintained. Please check and revise accordingly to the journal's instructions: https://www.mdpi.com/journal/metals/instructions

Author Response

Responds to reviewer2's comments:

The present manuscript entitled “1300 °C and 1400 °C Isotherms of SiO2-Al2O3-FeOx System in Air” by Song Li et al., describes the equilibrium phase relations for the key SiO2-Al2O3-FeOx system at 1300 °C and 1400 °C in air were experimentally determined by the equilibrium-quenching technique followed by XPS and SEM & EDX analysis. However, obvious discrepancies with about 20 °C were confirmed from further comparison with the predictions by FactSage were comprehensive conducted, to evaluate the need for updating the current thermodynamic oxide database related to metallurgical smelting. The authors report an interesting approach but the presentation of the work is clear. The objective and justification of the work are clear, and the experimental work is significant. The study is accurate and adequate, and thus, I would recommend it for publication in Metals. However, certain Minor issues are detailed below to improve the quality of the manuscript.

I advise the authors to take the following points into account while revising their manuscript.

Comment 1: There are some typographical errors in the manuscript text, so the authors need to correct them in the revised manuscript. In the whole manuscript, the authors must be taken care of the superscripts and subscripts, and abbreviations.

Response: Thanks for reviewer’s careful reading, the typographical errors, as well as the superscripts and subscripts, abbreviations have been revised in the whole manuscript, the revised part was marked as red in the manuscript.

Comment 2: Abstract is poorly written, it needs to be revised.

Response: Thanks for the suggestion, abstract was revised accordingly in the manuscript.

Comment 3: The authors need to add and discuss some more recent literature in the introduction section to strengthen the background of their work.

Response: Thanks for the careful advice, more recent references have been added to the introduction part to strengthen the discussion for the important application of phase diagram, the widely used metallurgical slag, and the research status of the current studied system.

Comment 4: The authors should explore and discuss the XPS results (Section 3.1.) with some more references to prepare a better discussion.

Response: In the revised manuscript, the mol ratio of Fe2+/Fe3+ was estimate as 0.87:1 according to the relative areas of the Gaussian peaks under Fe2+ and Fe3+ by the methods from reference [R1], the result indicate that the magnetite phase presented as solid solution phase properties, which is consistent with the information from Fe-O phase diagram. While for the consideration of convenient presentation, the stoichiometric composition of Fe3O4 was employed in the present study. The corresponding discussion has been revised accordingly in part 3.1 of the revised manuscript.

Reference:

R1. Liu R, Zhang Y, Duan L, et al. Effect of Fe2+/Fe3+ ratio on photocatalytic activities of Zn1-xFexO nanoparticles fabricated by the auto combustion method[J]. Ceramics International, 2020, 46(1): 1-7.

Comment 5: The homogeneity of the reference section needs to be maintained. Please check and revise accordingly to the journal's instructions: https://www.mdpi.com/journal/metals/instructions

Response: Thanks for the suggestion, the format of the reference part has been unified accordingly.

Reviewer 3 Report

In the thermodynamic equilibrium-quenching technique, you need to explain specific quenching conditions that might affect phase changes in your experiments.

Author Response

Responds to reviewer3's comments:

Comment: In the thermodynamic equilibrium-quenching technique, you need to explain specific quenching conditions that might affect phase changes in your experiments.

Response: Thanks for reviewer’s professional suggestion. The purpose of the quenching process is to maintain the equilibrium phase assembly and compositions to room temperature for convenient analysis, therefore, the basic requirement is that the temperature decrease speed in the quenching media should be faster than the speed of phase transformation that might happen during quenching process.

Based on reviewer’s comment, the specific quenching conditions have been added, and the manuscript has been revised accordingly in the experimental part.

Round 2

Reviewer 1 Report

NA